# WINNING THE L2RPN CHALLENGE: POWER GRID MANAGEMENT VIA SEMI-MARKOV AFTERSTATE ACTOR-CRITIC

**Deunsol Yoon**[*,1]**, Sunghoon Hong**[*,1]**, Byung-Jun Lee**[2,3]**, Kee-Eung Kim**[1,2]
[1]Graduate School of AI, KAIST, Daejeon, Republic of Korea
[2]School of Computing, KAIST, Daejeon, Republic of Korea
[3]Gauss Labs Inc., Seoul, Republic of Korea
{dsyoon,shhong,bjlee}@ai.kaist.ac.kr, kekim@kaist.ac.kr

## ABSTRACT

Safe and reliable electricity transmission in power grids is crucial for modern society. It is thus quite natural that there has been a growing interest in the automatic management of power grids, exemplified by the Learning to Run a Power Network Challenge (L2RPN), modeling the problem as a reinforcement learning (RL) task. However, it is highly challenging to manage a real-world scale power grid, mostly due to the massive scale of its state and action space. In this paper, we present an off-policy actor-critic approach that effectively tackles the unique challenges in power grid management by RL, adopting the hierarchical policy together with the afterstate representation. Our agent ranked first in the latest challenge (L2RPN WCCI 2020), being able to avoid disastrous situations while maintaining the highest level of operational efficiency in every test scenario. This paper provides a formal description of the algorithmic aspect of our approach, as well as further experimental studies on diverse power grids.

## 1 INTRODUCTION

The power grid, an interconnected network for delivering electricity from producers to consumers, has become an essential component of modern society. For a safe and reliable transmission of electricity, it is constantly monitored and managed by human experts in the control room. Therefore, there has been growing interest in automatically controlling and managing the power grid. As we make the transition to sustainable power sources such as solar, wind, and hydro (Rolnick et al., 2019), power grid management is becoming a very complex task beyond human expertise, calling for data-driven optimization.

Yet, automatic control of a large-scale power grid is a challenging task since it requires complex yet reliable decision-making. While most approaches have focused on controlling the generation or the load of electricity (Venkat et al., 2008; Zhao et al., 2014; Huang et al., 2020), managing the power grid through the topology control (changing the connection of power lines and bus assignments in substations) would be the ultimate goal. By reconfiguring the topology of the power grid, it can reroute the flow of electricity, which enables the transmission of electricity from the producers to consumers efficiently and thus prevent surplus production. There are preliminary studies of the grid topology control in the power systems literature (Fisher et al., 2008; Khodaei & Shahidehpour, 2010), but due to its large, combinatorial, and non-linear nature, these methods do not provide a practical solution to be deployed to the real-world.

On the other hand, deep Reinforcement Learning (RL) has shown significant progress in complex sequential decision-making tasks, such as Go (Silver et al., 2016) and arcade video games (Mnih et al., 2015), purely from data. RL is also perceived as a promising candidate to address the challenges of power grid management (Ernst et al., 2004; Dimeas & Hatziargyriou, 2010; Duan et al., 2020; Zhang et al., 2020; Hua et al., 2019). In this regard, we present Semi-Markov

---

[*] : Equal contribution

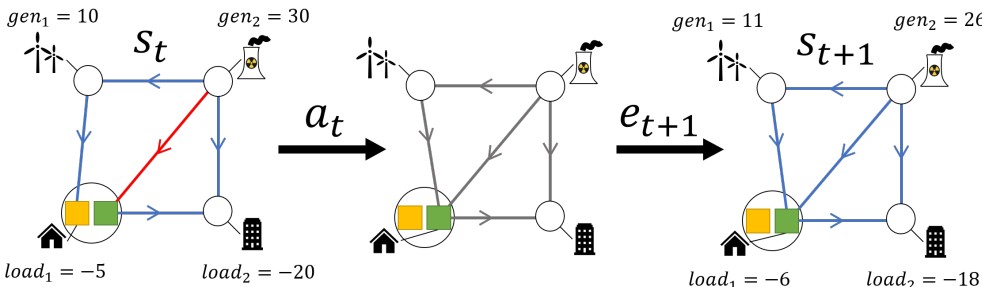

Figure 1: An example of a power grid with 4 substations, 2 generators, 2 loads, and 5 lines. Starting from the left, a bus assignment action $a_t$ reconfigures the grid and then the next state $s_{t+1}$ is determined by exogenous event $e_{t+1}$, such as the change of power demands in loads. The diagonal line was experiencing overflow, but the action $a_t$ is shown to revert the overflow. The power loss also reduced from 15 to 13.

Afterstate Actor-Critic (SMAAC), an RL algorithm that effectively tackles the challenges in power grid management.

One of the main challenges in RL for the real-world scale power grid management lies in its massive state and action space. We address the problem by adopting a goal-conditioned hierarchical policy with the afterstate representation. First, we represent state-action pairs as afterstates (Sutton & Barto, 2018), the state *after* the agent has made its decision but *before* the environment has responded, to efficiently cover the large state-action space. The afterstate representation can be much more succinct than the state-action pair representation when multiple state-action pairs are leading to an identical afterstate. For example, in the case of controlling the topology of the power grid, a pair of a current topology and an action of topology modification can be represented as a reconfigured topology, since the topology is deterministically reconfigured by the action. Then the next state is determined by random external factors, such as the change of power demands in load. Second, we extend this idea to a hierarchical framework, where the high-level policy produces a desirable topology under the current situation, and the low-level policy takes care of figuring out an appropriate sequence of primitive topology changes. Combined together, our hierarchical policy architecture with afterstates facilitates effective exploration for good topology during training.

Our algorithm ranked first in the latest international competition on training RL agents to manage power grids, Learning To Run a Power Network (L2RPN) WCCI 2020. In this paper, we further evaluate our approach using Grid2Op, the open-source power grid simulation platform used in the competition, by training and testing the agent in 3 different sizes of power grids. We show that the agent significantly outperforms all of the baselines in all grids except for the small grid where the task was easy for all algorithms.

## 2 BACKGROUND

### 2.1 GRID2OP ENVIRONMENT

We briefly overview Grid2Op, the open-source simulation platform for power grid operation used in the L2RPN WCCI 2020 challenge. Grid2Op models realistic concepts found in real-world operations used to test advanced control algorithms, which follow real-world power system operational constraints and distributions (Kelly et al., 2020).

The power grid is essentially a graph composed of nodes corresponding to *substations* that are connected to *loads*, *generators*, and *power lines*. The generator produces electricity, the load consumes electricity, and the power line transmits electricity between substations. The substation can be regarded as a router in the network, which determines where to transmit electricity. Grid2Op considers 2 conductors per substation, known as the double busbar system. This means that the elements connected to a substation, i.e. loads, generators, and power lines, can be assigned to one of the two busbars, and the power travels only over the elements on the same busbar. Thus, each substation can be regarded as being split into two nodes.

The state of the power grid consists of various features such as a topology configuration (the connectivity of each power line and the bus assignment in each substation), as well as the amount of power provided by each generator, required by each load, transmitted in each line, and so on. The power supplied by generators and demanded by loads changes over time, and the power transmitted in lines also changes according to the current topology configuration together with supply and demand. In addition, each line has its own capacity to transmit electricity and can be automatically disconnected when there is an overflow of electricity.

The agent can apply actions on substations and lines to managing the power grid. The action on a substation, called *bus assignment*, assigns the elements in the substation to a busbar. The action on a line, called *line switch*, disconnects (both ends of the line is assigned to neither bus) a line or reconnects a disconnected line. The agent is allowed to perform one line switch or one bus assignment action per step, and cannot successively perform actions on the same line or substation.

The power grid is simulated for a given period, typically for several days at a 5-minute interval. The simulation can terminate prematurely when the agent fails to manage the grid, i.e. (1) the amount of power required by loads are not delivered, which can happen if there are too many disconnected lines, or (2) a disconnected subgraph is formed as a result of applying an action. This is reflected in the failure penalty when measuring the performance of the agent, given by the number of remaining simulation time steps upon termination. Another important performance metric is the power loss penalty, given by the amount of power that disappeared during transmitting due to resistive loss. Thus, the goal of the agent is to operate the power grid both safely and efficiently by minimizing the failure penalty and the power loss penalty.

Figure 1 illustrates how the actions affect the state of the power grid using the bus assignment action as an example. The simulator provides 3 different sizes of power grids, (1) IEEE-5 is the power grid with 5 substations, (2) IEEE-14 is the power grid with 14 substations, and (3) L2RPN WCCI 2020 is the power grid with 36 substations. See Appendix A.1 for more details on the environment.

## 2.2 AFTERSTATES IN RL

Grid2Op provides a natural framework to use RL for operating power grids: we assume a Markov decision process (MDP) defined by $(S, \mathcal{A}, p, r, \gamma)$ to represent the RL task, where $S$ is the state space, $\mathcal{A}$ is the action space, and $p(s_{t+1}|s_t, a_t)$ is the (unknown) state transition probability, $r_t = r(s_t, a_t) \in \mathbb{R}$ is the immediate reward, and $\gamma \in (0, 1)$ is the discount factor. We assume learning a stochastic policy $\pi(a_t|s_t)$, which is a probability distribution over actions conditioned on states. The state and action value functions under $\pi$ are $V^\pi(s) = \mathbb{E}_\pi[\sum_{l \geq 0} \gamma^l r_{t+l}|s_t = s]$ and $Q^\pi(s, a) = \mathbb{E}_\pi[\sum_{l \geq 0} \gamma^l r_{t+l}|s_t = s, a_t = a]$ respectively.

As shown in Figure 1 in the previous section, the transition in Grid2Op comprises two steps: the topological change that results directly from the action, and then the rest of the state changes that arise from exogenous events. This motivates the use of the *afterstate* (Sutton & Barto, 2018), also known as the post-decision state in Approximate Dynamic Programming (ADP) (Powell, 2007), which refers to the state *after* the agent has made its decision but *before* the arrival of new information.

Let us define the state $S$ as $(\mathcal{T}, X)$ where $\mathcal{T}$ is the part of the state that is deterministically changed by an action, and $X$ as independent or affected indirectly from an action. Following the modeling in (Powell, 2007), the transition is decomposed into two parts using $f^A$ and $f^E$:

$$s_{t+1} = [\tau_{t+1}, x_{t+1}] = f^E([\tau_{t+1}, x_t], e_{t+1}), \quad s_t^{a_t} = [\tau_{t+1}, x_t] = f^A([\tau_t, x_t], a_t), \quad (1)$$

where $\tau_{t+1}$, the deterministic part of $s_{t+1}$, is given by the the function $f^A(s_t, a_t)$, and $x_{t+1}$, the stochastic part, is given by the function $f^E(s_t^a, e_{t+1})$ where $e_{t+1}$ is the source of the randomness in the transition sampled from some unknown distribution $p^E$. Note that $e_{t+1}$ itself can be included as a part in $x_{t+1}$.

Using the afterstate has a number of advantages. For example, if the state and the action spaces are very large but the set of unique afterstates is relatively small, learning the value function of afterstates would be much more efficient. The value of an afterstate $s^a$ under policy $\pi$ is defined as $V^\pi(s^a) = \mathbb{E}_\pi[\sum_{l \geq 0} \gamma^l r_{t+l}|s^a = f^A(s_t, a_t)]$ and its recursive form can be written as :

$$V^\pi(s_t^{a_t}) = \mathbb{E}_{e_{t+1} \sim p^E, a_{t+1} \sim \pi}\left[r(s_t, a_t) + \gamma V^\pi(f^A(s_{t+1}, a_{t+1}))|s_{t+1} = f^E(s_t^{a_t}, e_{t+1})\right] \quad (2)$$

The optimal afterstate value function and the optimal policy can be obtained by iteratively alternating between the policy evaluation by Eq. (2) and policy improvement :

$$\pi_{new}(s_t) = \arg\max_{a_t} \left[ V^{\pi_{old}} \left( f^A(s_t, a_t) \right) \right] \tag{3}$$

Note that we cannot gain much from the afterstate representation when using the individual power grid operations as actions since they result in unique changes in the grid topology. However, we shall see that the afterstate becomes very powerful when we consider the sequences of grid operations as the action space, where their permutations result in identical changes in the final topology.

## 3 APPROACH

We first present the state space, the action space, and the reward function modeled in our approach. Then we briefly explain the unique challenge in Grid2Op and describe our approach to tackle the challenge. Finally, we will describe the overall architecture of the RL agent.

### 3.1 MODELING STATES, ACTIONS AND REWARDS

**State** We also define the state $S$ in the Grid2Op environment as $(\mathcal{T}, X)$ where $\mathcal{T}$ is set of topology configuration (deterministically changed by action) and $X$ as various features as power demands and supplies (independent of the action), power being transmitted in each line (affected indirectly from the action) and so on. The detail about the features of states used in this work is provided in Appendix A.1.

**Action** We only consider bus assignment actions in our agent: we assume that it is desirable to have as many lines connected as possible since the overflow is less likely to occur when there are many routes for the power delivery. Thus, for line switch actions, we simply follow the rule of always reconnecting the power lines whenever they get disconnected due to the overflow.

Let us define the number of the substation as $N_{sub}$ and elements in $i$th substation as $Sub(i)$. Each end of lines, generators, and loads in the substation can be assigned to one of two busbars, so the total number of actions is $|\mathcal{A}| = \sum_{i=0}^{N_{sub}} 2^{Sub(i)}$ (i.e. each action chooses one of the substations and perform a bus assignment therein). Following the approach taken by the winner of the previous challenge L2RPN 2019 (Lan et al., 2019), we made our agent act (i.e. intervene) only in hazardous situations. The condition for being hazardous is determined by the existence of a line in which the power flow is larger than the threshold hyperparameter. This naturally yields a semi-MDP setting for RL (Sutton et al., 1999).

**Reward** We define the reward in intermediate time steps to be the efficiency of the power grid, defined by the ratio of the total load to the total production, i.e. $\frac{load_t}{prod_t}$. Note that if the ratio becomes greater than 1, the episode terminates with a large penalty for the failure since the production does not meet the demand.

### 3.2 ACTOR-CRITIC ALGORITHM WITH AFTERSTATES

The main challenge of the Grid2Op environment is the large state and action spaces. For the power grid with 36 substations used in the L2RPN WCCI 2020 competition, there are about 70,000 actions that yield unique changes to the topology. We address this problem by adopting the actor-critic architecture, where the policy and the value function are represented by function approximators. In addition, we use the afterstate representation to capture many state-action pairs being led to an identical afterstate by leveraging the transition structure, shown in Figure 1. For notational simplicity, all the derivations assume MDP in this section, which shall be extended to the semi-MDP setting in the next section.

We use function approximators for the afterstate value function $V_\psi(s_t^{a_t})$ and policy $\pi_\theta(a_t|s_t)$ parameterized by $\psi$ and $\theta$ respectively. The actor is trained to maximize $J_\pi$ and the critic to minimize $L_V$ :

$$J_\pi(\theta) = \mathbb{E}_{s_t \sim D, a_t \sim \pi_\theta(\cdot|s_t)} \left[ V_\psi(f^A(s_t, a_t)) \right] \tag{4}$$

$$L_V(\psi) = \mathbb{E}_{(s_t^{a_t}, s_{t+1}) \sim D} \left[ \left( V_\psi(s_t^{a_t}) - r(s_t, a_t) - \gamma \mathbb{E}_{a_{t+1} \sim \pi_\theta(\cdot | s_{t+1})} \left[ V_\psi(f^A(s_{t+1}, a_{t+1})) \right] \right)^2 \right] \quad (5)$$

where the replay buffer $D$ stores the transition tuple $[s_t, s_t^{a_t}, r(s_t, a_t), s_{t+1}]$ for off-policy learning. The actor and the critic are trained using Soft Actor-Critic (SAC) (Haarnoja et al., 2018). Note that it learns a value function over an afterstate with a reconfigured topology, rather than a state-action pair, which is more succinct. Although the above equation defines a state-value critic, we can still train off-policy since it is essentially an action-value critic (i.e. an afterstate is defined by a state and an action).

Furthermore, we aim to apply the gradient estimator through a reparameterization trick similar to Haarnoja et al. (2018), since it is known to have lower variance than the likelihood ratio gradient estimator, resulting in stable learning. In order to update the actor via reparameterization trick, the transition $f^A$ must be differentiable, but it is not straightforward to define $f^A$, which maps from the bus assignment actions to the topology configurations, as a differentiable formula. In the next section, we will mitigate the problem by re-defining the action space.

## 3.3 EXTENSION TO GOAL-CONDITIONED HIERARCHICAL FRAMEWORK

It is very challenging to take exploratory actions in the Grid2Op environment: if the agent takes random actions, the power grid would fail in a few time steps. For example, the agent with the random policy would mostly fail in less than 10 time steps, whereas the agent with the no-op policy (naively maintaining the initial grid topology throughout time steps) would survive approximately 500 time steps on average. Thus, it is very difficult for the agent to explore diverse grid topology configurations that are significantly different from the initial ones, and thereby the random exploration policy (e.g. $\epsilon$-greedy) would be often stuck at bad local optima that executes only one or two actions. Therefore, a more structured exploration is a key to successful training.

To this end, we extend the afterstate actor-critic algorithm to a two-level hierarchical decision model by defining the goal topology configuration as the high-level action. Specifically, we define the high-level actions as the goal topology configuration $g \in \{0,1\}^n$ where $n = \sum_{i=0}^{N_{sub}} Sub(i)$, which is learned by the high-level policy $\pi^h$. This leads to the temporally extended afterstate representation, given by $s_t^{g_t} = [\tau_{t+d} = g_t, x_t] = f^A([\tau_t, x_t], g_t)$ where $t$ denotes the time a hazard occurs and $d$ denotes the time interval next hazard occurs. Note that we can now take full advantage of the afterstate representation since the equivalence of many different sequences of primitive actions (i.e. individual bus assignment actions) that lead to the identical topology are now captured by the goal topology configuration.

In addition, exploration with goal topology is more effective than with primitive actions since the policy only needs to focus on *where to go*, i.e. the desirable topology under the current situation, without needing to care about *how to get there*, i.e. figuring out a suitable primitive action sequence that would yield the goal topology, with the help from an appropriate low-level policy. Finally, we can now use the reparameterization trick for the actor update in a straightforward manner since the result of $f^A$ is merely a copy of the action $g_t$.

The replay buffer $D$ stores the transition tuple, $[s_t, g_t, r_{t:t+d}, s_{t+d}]$ where $r_{t:t+d} = \sum_{t'=t}^{t+d} \gamma^{t'-t} r_{t'}$. The high-level policy can be trained through the objective function of the actor and the critic written as:

$$J_\pi(\theta) = \mathbb{E}_{g_t \sim \pi_\theta^h} \left[ V_\psi([g_t, x_t]) \right] \quad (6)$$

$$J_V(\psi) = \mathbb{E}_D \left[ \left( V_\psi(s_t^{g_t}) - r_{t:t+d} - \gamma^d \mathbb{E}_{g_{t+d} \sim \pi_\theta^h} \left[ V_\psi([g_{t+d}, x_{t+d}]) \right] \right)^2 \right] \quad (7)$$

As for the low-level policy, it is relatively simple to find the action sequence that changes the current topology into the goal topology: we just need to identify the set of substations that requires changes in the bus assignment and make appropriate reassignments therein. Thus, we take a rule-based approach for the low-level policy, $a_t = \pi_{rule}^l(s_t, g_t)$ where the $rule$ determines the order of substations to execute bus assignment actions. For example, we could impose a priority on substations such that the substations with the least room in the capacity make their bus reassignment first because they are the ones requiring the most urgent interventions. In the experiments section, we compare the results using various rules including a learning-based approach.

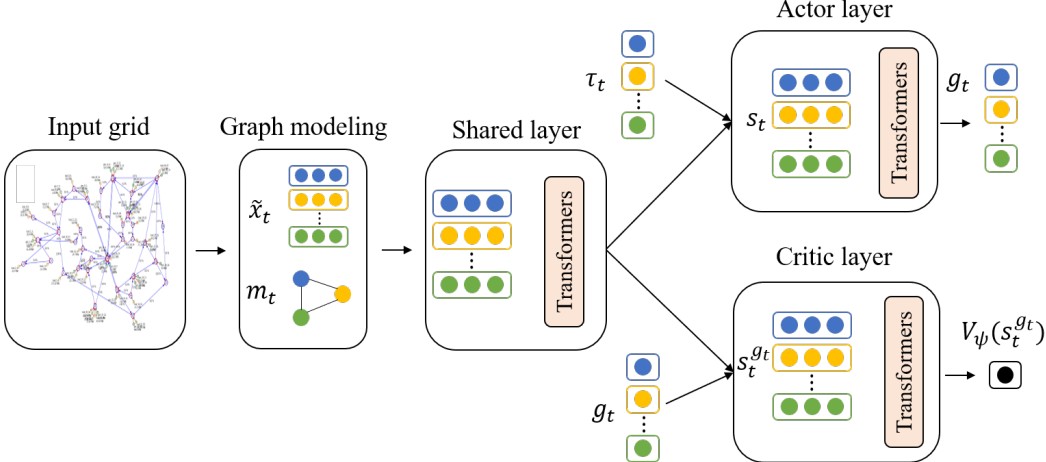

Figure 2: Overview of our model. The shared layer encodes $x_t$, the actor layer outputs the desirable topology $g_t$ given the current state $s_t = [\tau_t, x_t]$, and the critic layer outputs the afterstate value given the afterstate $s_t^{g_t} = [g_t, x_t]$.

### 3.4 IMPLEMENTATION

In order to leverage the interconnection structure of the power grid, we apply graph neural networks (GNN) (Scarselli et al., 2008). As illustrated in Figure 2, given the power grid with $n$ substations, we reshape $x_t$ in the state $s_t = [\tau_t, x_t]$, given as a flat vector in Grid2Op, into $(M, \tilde{x}_t)$, where $M \in \{0, 1\}^{n \times n}$ is the adjacency matrix, and $\tilde{x}_t \in \mathbb{R}^{n \times k}$ is the node matrix with $k$ features. We adopted the transformer (Vaswani et al., 2017) as the GNN block, where the adjacency matrix $M$ is used for masking out the attention weights of nodes, following the architecture proposed by Parisotto et al. (2020). The actor and the critic share the lower layers, consisting of GNN blocks and linear layers. Additionally, we add an entropy of policy to the objective function of the actor and the critic, following the SAC formulation. Details of the architecture are provided in Appendix A.2.

## 4 RELATED WORKS

The topology control of the power grid through line switch has been previously studied in the power systems literature. Previous works, Fisher et al. (2008) and Khodaei & Shahidehpour (2010), solve the optimal transmission switching problem by mixed-integer programming. Since then, several heuristics have been introduced to tackle the computational cost (Fuller et al., 2012; Dehghanian et al., 2015; Alhazmi et al., 2019). Recently, Marot et al. (2018) explores bus assignment, more complex than the line switch, and presents an algorithm based on expert knowledge, which shows the utility of bus assignment. Their algorithm can find remedial bus assignment action that can revert overflow with a high probability of success and acceptable computational time. Han (2020) explores bus and line separation to solve the problem of short circuit current reduction in power systems through RL. Marot et al. (2020) models the power grid management through line switch and bus assignment as a RL task and releases an open-source simulation called Grid2Op for power grid management in multi-step time horizons. Additionally, they held the international power grid management competition, L2RPN 2019 challenge, where IEEE-14 is chosen for the competition environment, and Subramanian et al. (2020) present a simple deep RL approach for IEEE-14.

The winner of the L2RPN 2019 challenge (Lan et al., 2019) tackles the problem through pre-training and guided exploration. They collect massive data sets from the simulator which can restore particular states, and pre-train an agent to generate a good initial policy. For exploration in the large action space, they use guided exploration instead of random exploration, where the agent simulates the top few actions with high action values before performs its action to the environment at every time step. They also design the agent to act only in hazardous situations, and they train it using dueling Deep Q-Networks (DQN) (Wang et al., 2016) and prioritized replay buffer (Schaul et al., 2016).

| Grid | $N_{sub}$ | $N_{line}$ | $N_{gen}$ | $N_{load}$ | $|S|$ | $|\mathcal{A}|$ | $n$ |
|---|---|---|---|---|---|---|---|
| IEEE-5 | 5 | 8 | 2 | 3 | 74 | 58 | 21 |
| IEEE-14 | 14 | 20 | 6 | 11 | 194 | 160 | 57 |
| L2RPN WCCI 2020 | 36 | 59 | 22 | 37 | 590 | 66810 | 177 |

Table 1: Characteristics of the grids. $N_{sub}$, $N_{line}$, $N_{gen}$, $N_{load}$ is the total number of substations, lines, generators, and loads. $|S|$ is the dimension of state, $|\mathcal{A}|$ is the number of unitary bus assignment actions, $n$ is the dimension of topology configuration.

The afterstate representation has been applied to address the resource allocation problems and dynamic routing problems. Singh & Bertsekas (1996) formulate the dynamic channel allocation problem in the cellular network as a dynamic programming problem using the afterstate value function. More recently, there has been research on utilizing the afterstate representation combined with ADP in a dynamic vehicle routing problem (Agussurja et al., 2019; Ulmer et al., 2019). Shah et al. (2020) also apply an afterstate-based deep RL method in a ride-pool matching problem. The hierarchical framework has long held the promise to tackle complex RL tasks (Dayan & Hinton, 1992; Parr & Russell, 1998; Barto & Mahadevan, 2003), and especially one of the prevailing approaches, the goal-conditioned hierarchical framework has recently achieved significant success in various tasks, such as simulated and real-world quadrupedal manipulation (Nachum et al., 2018a; 2020) and complex navigation (Levy et al., 2019; Zhang et al., 2020). However, to the best of our knowledge, none of the works combines the afterstate representation with a hierarchical framework.

GNN has been widely used in deep RL to directly tackle graph-structured problems or to represent the interaction between entities in a state. You et al. (2018) formulates goal-directed graph generation as MDP and solves designing a molecular structure with specific desired properties problem through an RL algorithm. Wang et al. (2018) apply GNN for continuous control by modeling controllable joints as nodes for a graph, and the physical dependencies between joints as edges to capture underlying graph structure. Zambaldi et al. (2019) adopt GNN for a navigation and planning task where complex relational reasoning is required to represent pairwise interactions between objects in a state, and Jiang et al. (2020) adopt it for learning cooperation in multi-agent environments by modeling agents as nodes in a graph where they communicate through GNN.

## 5 EXPERIMENTS

### 5.1 EXPERIMENTAL SETUP

Our experiments are conducted on the 3 power grids, IEEE-5 (smallest), IEEE-14, and L2RPN WCCI 2020 (largest, used in the challenge), provided by Grid2Op. Details of each grid are provided in Table 1. Each grid has a set of scenarios, and each scenario specifies the variations in the simulation such as the power supplies and demands at each time step. The length of each scenario is 864 time steps, which corresponds to 3 days at 5 minute time-resolution.

Since Grid2Op is relatively new to the research community, there are few RL methods applied to the grid topology control. Therefore we implement 3 baselines for performance comparison to verify the effectiveness of our method: (1) **DDQN** (Dueling DQN) has similar architecture as the last winner of the challenge, which learns the action-value function with the primitive action space (2) **SAC** is similar to DDQN but utilizes maximum entropy exploration following SAC algorithm. (3) **SMAAC\AS** is SMAAC without the afterstate representation, where we use *action*-value critic $Q^\pi(s, g)$. Thus, DDQN and SAC assume the MDP setting with primitive actions, SMAAC\AS assumes the goal-conditioned semi-MDP setting but without the afterstate representation. We additionally compare our approach with the 3rd placed participant in the L2RPN WCCI 2020 grid,[1] (4) **YZM**, the only agent with publicly available code. This agent heuristically selects 596 actions among the primitive actions in prior and trains the agent with the reduced action space using Asynchronous Advantage Actor-Critic (Mnih et al., 2016). YZM additionally trains a backup agent

---

[1]Their algorithm is designed specifically for L2RPN WCCI 2020 grid and cannot be applied to other grids.

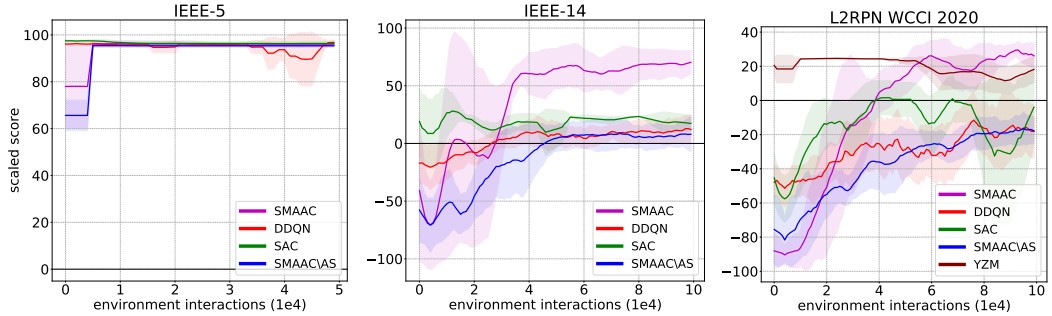

Figure 3: Training curves on 3 grids. Evaluation rollouts are performed every 1000 steps, and the shaded area represents the standard error.

|  | IEEE-5 | IEEE-14 | L2RPN WCCI 2020 |
|---|---|---|---|
| **SMAAC** | 98.18± 0.31 | **69.66 ± 10.62** | **55.26 ± 5.82** |
| DDQN | 97.66 ± 1.04 | 29.11 ± 16.00 | 26.22 ± 7.39 |
| SAC | 98.26 ± 0.06 | 43.93 ± 0.02 | 39.1 ± 2.94 |
| SMAAC\AS | 93.79 ± 0.58 | 14.91 ± 14.08 | 11.58 ± 2.74 |
| YZM | - | - | 28.34 ± 4.31 |

Table 2: Performance measured by the scaled score on the 10 test scenarios averaged over 3 instances with standard error. Each the best policy is obtained from the one with the highest performance in the validation scenarios during training.

with another set of actions and invokes the backup agent when the base agent can lead to overflow or termination by using the simulation function.[2]

For a fair comparison, all baselines except for YZM encode the input state through the same GNN architecture, and the agents get activated only in hazardous situations. The detail of implementation is provided in Appendix A.3 and the code is provided in https://github.com/sunghoonhong/SMAAC.

### 5.2 RESULTS

Figure 3 shows the total average scaled score of evaluation rollouts on the 10 validation scenario set during training: the scores are scaled in the range [-100,100], with the return of the no-op agent scaled and translated to 0, indicating how better the agent manages the power grid than the no-op agent in terms of safety and power efficiency. Each algorithm was trained and evaluated for 3 runs for averaging the scores.

As shown in Figure 3, all algorithms easily solve the smallest grid (IEEE-5). In the medium (IEEE-14) and the large (L2RPN WCCI 2020) grids, both DDQN and SAC perform poorly. DDQN performs slightly better than the no-op agent in the medium grid and worse than the no-op agent in the largest grid. Exploring with primitive actions is extremely difficult since most actions can lead to disastrous termination, and thereby it cannot find grids other than the initial one. This yields the DDQN to be stuck at bad local optima, not much better than the no-op agent. SAC performs slightly better than DDQN in the larger grids. This is due to the sophisticated optimization scheme in SAC that is shown to affect a number of other RL benchmark tasks. However, in Grid2Op, the performance was barely better than the no-op agent due to the same challenge faced by DDQN.

Perhaps surprisingly, the performance of SMAAC\AS is no better than using primitive actions, although the hierarchical decision encourages deviating from the initial topology. Without the afterstate representation, the critic was not able to learn a good action-value function due to the massive state and action spaces. YZM uniquely leverages the simulation function and can show good performance from the beginning. However, exploring primitive actions is still hard even with

---

[2]The simulation function is a predefined method in Grid2Op, which returns a next state given an action through an approximate simulation.

|  | **1 (ours)** | 2 | 3 | 4 | 5 | 6 | 7 |
|---|---|---|---|---|---|---|---|
| Scaled score | 75.72 | 66.21 | 48.62 | 26.60 | 17.98 | 4.31 | 0.07 |
| CPU time (sec) | 812.49 | 1406.45 | 1233.08 | 1322.02 | 116.43 | 96.56 | 118.58 |

Table 3: The top 7 leaderboard of the L2RPN WCCI 2020 Challenge among 50 participants.

the reduced set of actions, and it can be observed that YZM struggles to improve its performance. The performance on the test scenarios is provided in Table 2. We provide a qualitative analysis of how each agent behaves differently and how SMAAC remedies the hazardous power grid with a detailed example in Appendix A.4.

On the contrary, our method learns significantly fast and outperforms all the baselines, effectively combining the benefits of the hierarchical decision model and the afterstate representation. Finally, Table 3 shows the leaderboard in the L2RPN WCCI 2020 challenge.

### 5.3 LOW-LEVEL RULE DESIGN

In this section, we examine how the low-level policy affects the overall performance. (1) **FIXED** gives priority to substations randomly that are predefined and fixed during training. We implement this low-level agent to find out whether our high-level agent can manage the power network on the poor low-level agent. (2) **CAPA** gives high priority to substations with lines under high utilization of their capacity, which applies an action to substations that require urgent care. (3) **DESC** imposes a priority on large substations, i.e. many connected elements. A change to a large substation can be seen as making a large change in the overall topology with a single action. (4) **OPTI** optimizes execution order by training, making the actor additionally output $N_{sub}$ values that represents the priority of substations. All rules achieve similar performance with overlapped confidence intervals except for FIXED.

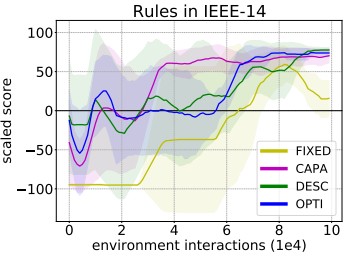

Figure 4: Comparison of 4 rules.

As shown in Figure 4, especially, CAPA converges fast compared to OPTI and DESC, hence we use this low-level agent in Section 5.2. We assume that most of the rules could achieve similar final performances since SMAAC is resilient to suboptimal low-level rules. By generating subgoals that include a subset of intended topology reconfiguration, the high-level policy can adapt to suboptimal low-level rules to form an optimal policy overall. However, as the result of FIXED suggests, a very poorly designed low-level policy can lead to instability and degrade the performance.

## 6 CONCLUSION

In this paper, we presented SMAAC, a deep RL approach demonstrated to be very effective for power grid management. SMAAC is an actor-critic algorithm that combines the afterstate representation with a hierarchical decision model. This is very important for power grid management modeled by Grid2Op, where actions are too primitive for effective exploration and many permutations of action sequences lead to identical changes in the power grid topology. Besides, naive explorations with primitive actions are subject to immediate failure due to the unique nature of power grid management. We empirically demonstrated that the presented method significantly outperforms several baselines in the real-world scale power grids, and ranked first in the latest international competition, L2RPN WCCI 2020 challenge. Our work shows the possibility of an intelligent agent that automatically operates the power grid for several days without expert help.

### ACKNOWLEDGMENTS

This work was supported by the National Research Foundation (NRF) of Korea (NRF-2019M3F2A1072238 and NRF-2019R1A2C1087634), and the Ministry of Science and Information communication Technology (MSIT) of Korea (IITP No. 2019-0-00075, IITP No. 2020-0-00940 and IITP No. 2017-0-01779 XAI).

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

# A  APPENDIX

## A.1  ENVIRONMENT DETAIL

Grid2Op provides a simulation for power grid operation in real-time over several days at a 5-minute time-resolution. There are 3 power grids, IEEE-5, IEEE-14, and L2RPN WCCI 2020 (a subgraph of IEEE-118), where each has different size.

**State Space**   The state of the power grid consists of 12 features presented in Table 4. We use 5 features provided by the environment and 1 feature defined by us, which we consider enough to represent the current state of the grid; Active power, rho, topology configuration, time step overflow, maintenance, hazard. The maintenance is a boolean vector representing whether a line is in maintenance, and the hazard represent is also a boolean vector representing whether electricity flows of a line is larger than a predefined threshold $\delta_h$. We use 0.9 for the threshold, and this threshold is same as the one we used for the hazardous state. Further details are provided in `grid2op.readthedocs.io/en/latest/observation.html`

**Action Space**   The agent can apply actions on substations and lines to manage the power grid. The action on a substation, called *bus assignment*, assigns the elements in the substation to a busbar. The action on a line, called *line switch*, disconnects (both ends of line is assigned to neither bus) a line or reconnects a disconnected line. Let us define the number of lines in the power grid as $N_{line}$, the number of substations as $N_{sub}$, and the number of elements in $i$th substation as $Sub(i)$. Then at each time step the agent selects an action $a_t$ from the action space $\mathcal{A}$ where $|\mathcal{A}| = N_{line} + 2^2 \times N_{line} + \sum_{i=0}^{N_{sub}} 2^{Sub(i)}$. [3]

**Rule**   There are some rules that make the task more realistic and challenging in Grid2Op. Lines can be automatically disconnected due to overflow of current, i.e. if more current flows than a line can hold for 3 time steps, a line is automatically disconnected. There is cooldown time for each component, i.e. the agent cannot apply its action to the same component successively and it is reactivated after 3 time steps later. There is a stochastic event called maintenance that happens intermittently. During maintenance, a line is disconnected by force and it cannot be reconnected.

**Score**   The performance of an agent can be evaluated by a score which consists of power loss penalty, failure penalty, and redispatching penalty. It is defined as :

$$Score = \sum_{t=0}^{t_{over}} (prod_t - load_t) + \sum_{t=t_{over}}^{t_{end}} penalty + \sum_{t=0}^{t_{over}} redispatch_t \qquad (8)$$

where $t_{over}$ is the time game over occurs, $prod_t$ is total amount of power supply by all generators, $load_t$ is total amount of power demand by loads. $prod_t - load_t$ stands for total amount of power loss. The failure penalty is given by the sum over the large constant $penalty$ for the remaining simulation time steps upon termination. The redispatching penalty is incurred when the agent do redispatching action, which controls generators to produce more or less electricity. Since redispatching action always incurs the additional penalty, we do not consider this action, so the redispatching penalty is always 0 in our case. Therefore, the goal of the agent is to operate the power grid both safely and efficiently by minimizing the failure penalty and the power loss penalty.

## A.2  MODEL ARCHITECTURE

Given the state $s = [\tau, x]$, we reshape $x$ into $(M, \tilde{x})$, where $M \in \{0, 1\}^{n \times n}$ is the adjacency matrix, and $\tilde{x} \in \mathbb{R}^{n \times k}$ is the node matrix with $k$ features. The shared layers consisted of $L_s$ GNN blocks that computes the node embedding of an input graph through transformer layers at their beginning. Given input nodes matrix $\tilde{x}$, a linear layer with ReLU activation increases input dimension $k$ to embedding dimension $k_s$, which maps $\tilde{x} \in \mathbb{R}^{n \times d}$ to $H^0 \in \mathbb{R}^{n \times k_s}$. After the linear layer, $L_s$

---

[3]Both ends of the line can be assigned to one of two buses in the substation, so $2^2 \times N_{line}$ is the number of reconnection. Each end of the line, generator, and load in the substation also can be assigned to one of two buses in the substation, so bus switching actions are $\sum_{i=0}^{N_{sub}} 2^{Sub(i)}$

| Name | Type | Size | Description |
|------|------|------|-------------|
| Date | int | 6 | The current year, month, day, hour of day, minute of hour, day of week. |
| Active power | float | $N_{gen} + N_{load}$ $+2 \times N_{line}$ | Active power magnitude. |
| Reactive power | float | $N_{gen} + N_{load}$ $+2 \times N_{line}$ | Reactive power magnitude. |
| Voltage | float | $N_{gen} + N_{load}$ $+2 \times N_{line}$ | Voltage magnitude. |
| Rho | float | $N_{line}$ | The capacity of each power line, which is defined as ratio between current flow and thermal limit. |
| Topology Configuration | int | $N_{gen} + N_{load}$ $+2 \times N_{line}$ | For each element (load, generator, ends of a line), it gives on which bus these elements is connected in its substation. |
| Line status | bool | $N_{line}$ | The status of each line, whether it is connected or disconnected. |
| Time step overflow | int | $N_{line}$ | The number of time steps each line is overflowed. |
| Time before cooldown line | int | $N_{line}$ | How much cooldown time for each line is left. An agent cannot perform its action on lines with cooldown. |
| Time before cooldown sub | int | $N_{sub}$ | How much cooldown time for each substation is left. An agent cannot perform its action on substations with cooldown. |
| Time next maintenance | int | $N_{line}$ | The time of the planned maintenance. An agent cannot act on the lines with maintenance. |
| Duration next maintenance | int | $N_{line}$ | The number of time steps that the maintenance will last. |

Table 4: Details of features in the state

transformer layers follow. The input of the transformer block at the $l^{th}$ block is an embedding from the previous layer $H^{l-1}$ and the adjacency matrix $\mathcal{A}$, $H^l = Transformer(H^{l-1}, A)$.

The actor's head consists of $L_a$ transformer blocks and 2 linear layers. Given the final node embedding $H^{L_e}$ from the shared layers, the transformer layers in actor's head takes it as the input and outputs node embedding $H^{L_a} \in \mathbb{R}^{n \times k_a}$. The first linear layer transforms 2D node embedding $H^{L_a}$ to a vector node embedding $\mathbb{R}^n$ by reducing the embedding dimension $k_a$ to 1, which is then concatenated with the current topology $\tau$ to form the state $s$ and the next linear layers outputs mean and standard deviation of the normal distribution. We sample continuous values $g' \in \mathbb{R}^n$ from the normal distribution followed by tanh non-linearity and the desirable topology $g \in \{0,1\}^n$ is constructed by assigning 1 to values in $g'$ larger than predefined topology threshold $\delta_\tau$ and 0 otherwise. We empirically find out that an agent without the threshold has difficulty learning in the large grid. However, an appropriate threshold helps stable learning and fast convergence.

The critic's head has a similar structure except for linear layers. There are $L_c$ GNN blocks in the critic that also takes $H^{L_e}$ and outputs $H^{L_c} \in \mathbb{R}^{n \times k_c}$. After transforming it into a vector node embedding $\mathbb{R}^n$ by a linear layer, $g'$ is concatenated to $H^{L_c}$, and the following two linear layers take it as the input and outputs a scalar value. The overall architecture is shown in the Figure 2.

Although the agent observes all substations, we reduce the goal dimension $n$ to $\tilde{n}$ by restricting controllable substation. As we mentioned in subsection 3.1, we do not consider disconnection or reconnection. Therefore, the agent only controls substations that have more than 2 elements since there are only two possible cases, elements on a same bus (connection) or elements on a different bus (disconnection). For L2RPN WCCI 2020 grid, the agent acts on substations that have more than 5 elements in order for fast convergence. As a result, the goal dimension is reduced from 21 to 16 in IEEE-5, from 57 to 42 in IEEE-14, and from 177 to 79 in L2RPN WCCI 2020.

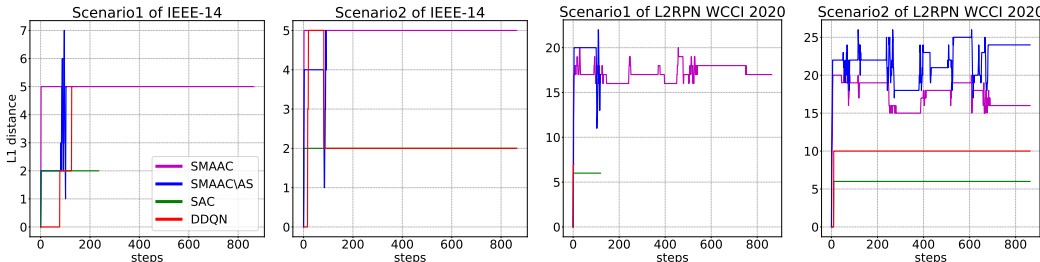

Figure 5: Deviation from the initial grid. We measure L1 distance between the initial topology and the current topology during rollout in 2 test scenarios for each grids.

## A.3 IMPLEMENTATION DETAILS

For all models, we use 6 state features, active power, rho, topology configuration, time step overflow, maintenance, and hazard, where $\delta_h = 0.9$. All the agent acts only when there is a rho of a line, ratio between current flow and thermal limit that is capacity of a line, is larger than the $\delta_h = 0.9$. Additionally the last 6 states of a history are stacked to represent the input state, since the difference between the state at time step $t$ and $t + 1$ is not significantly different in Grid2Op. Since the first decimal place of reward does not change significantly ($\frac{load_t}{prod_t}$ varies from 0.85 to 0.99 most of time), we transform it as $(\frac{load_t}{prod_t} \times 10 - 9) \times 0.1$ to use the second decimal place. Adam optimizer (Kingma & Ba, 2015) is used for training with $5e-5$ learning rate and 128 batch size. We perform grid search to find the best hyperparameters for each model.

**SMAAC** SMAAC is our proposed model, which learns the afterstate value function on the goal space, namely topology configuration space. We use $L_s = 6$ GNN blocks with embedding dimension $k_s = 64/128$ for shared layers. For actor's head, we use $L_a = 3$ GNN blocks with embedding dimension $k_a = 64/128$. For critic's head, we use $L_c = 1$ GNN block with embedding dimension $k_c = 64/128$ followed by linear layers with $\frac{k_c + \tilde{n}}{4}$ hidden units. We use $\delta_\tau = 0/0.1/0.15$. In practice, the reward for the high-level policy in Equation 6 is not discounted, $r_{t:t+d} = \sum_{t'=t}^{t+d} r_{t'}$ and $\gamma$ is used instead of $\gamma^d$ following Nachum et al. (2018b). For the competition, we use $k_s = 128$, $k_c = 128$, $k_a = 128$, $\delta_\tau = 0.35$, and $\tau$ is used as an extra input feature for the shared layers.

**SMAAC\AS** SMAAC\AS is a baseline, which learns the action-value function of desired relative change in hierarchical framework. The overall architecture is similar to SMAAC, but the critic takes both desired relative change and topology configuration to learn on state-action pairs. We use $\delta_\tau = 0/0.1/0.15$.

**SAC** SAC is a baseline, which learns the action-value function on the primitive bus assignment action space. We use $L_s = 6$ GNN blocks with embedding dimension $k_s = 64/128$ for shared layers. For actor's head, we use $L_a = 3$ GNN blocks with embedding dimension $k_a = 64/128$. And, we use softmax to output categorical distribution while utilize relaxed categorical distribution in training. For critic's head, we use $L_c = 1$ GNN block with embedding dimension $k_c = 32/64/128$ followed by concatenation with one-hot encoded action and one linear layer.

**DDQN** DDQN is a baseline, which learns the action-value function on the primitive bus assignment action space. It does not have separate actor but critic outputs $|\mathcal{A}|$ action-values. We use $L_s = 6$ GNN blocks with embedding dimension $k_s = 64/128$ for embedding layers. Following it, the critic has $L_c = 1$ GNN blocks with embedding dimension $k_s = 64/128$. Then, it utilizes the technique used by dueling DQN, namely we compute action-values through both value network and advantage network.

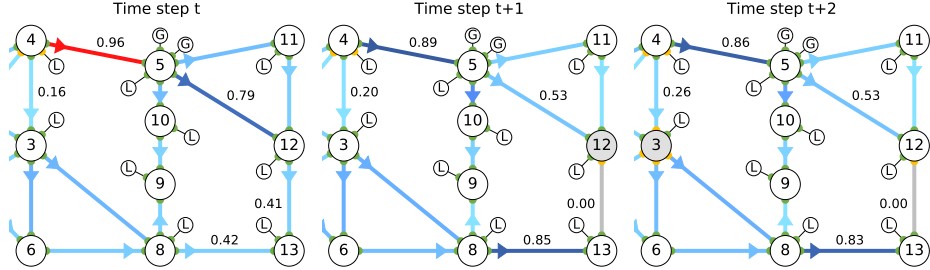

Figure 6: Illustration of a part of IEEE-14 managed by our agent. Large circle refers to a substation, small circle with G or L refers to a generator or load, and dots in green or yellow on a substation refers 2 busbars. The numbers beside lines are line usage ratio of current to line capacity and the darker color of the line means large ratio.

| Threshold $\delta_h$ | Average scaled score |
|---|---|
| 1.1 | $46.67 \pm 5.92$ |
| 1.0 | $52.11 \pm 6.91$ |
| 0.9 | $55.26 \pm 5.82$ |
| 0.8 | $36.08 \pm 16.53$ |

Table 5: Performance measured by the scaled score on the 10 test scenarios of L2RPN WCCI 2020 averaged over 3 instances with standard error. Each the best policy is obtained from the one with the highest performance in the validation scenarios during training.

## A.4 QUALITATIVE ANALYSIS

In this section, we present qualitative analysis based on example behaviors of agents. Figure 5 indirectly shows how each agent behaves in 2 grids. In the medium grid, after our agent reaches to the certain grid, it keeps staying in that grid. We speculate that the agent finds the optimal grid during training where electricity could distribute evenly all the time so no further actions required. It is reasonable in Grid2Op where a single action can potentially destroy the grid. On the other hand, it shows different behavior in the large grid. It changes the topology configuration diversely to revert the hazardous situations since the agent cannot find the grid such as the one in the medium grid.

SAC, which shows the best performance among the baselines, only changes a few of the topology configuration by executing only one or two actions in both grids, since the initial topology is strong local optima in Grid2Op. DDQN also shows similar behavior, but it shows worse performance than the SAC, since it changes the initial grid more. Likewise, SMAAC\AS that changes the topology the most shows the worst performance. It is extremely difficult to find the better topology than the initial one in Grid2Op without the effective exploration. The efficient learning together with the effective exploration is the key to successful management.

We further examine how SMAAC learned to revert a hazardous state back to the safe state. As shown in Figure 6, the line 4-5 (between substation 4 and 5) in $t$ is in the hazardous situation. At time step $t+1$, our agent makes bus assignment change in the substation 12 by assigning line 12-13 to yellow busbar. As a result, some amount of electricity that flows in the line 4-5 moves to line 4-3 to meet the load demand in the substation 13 where electricity is supplied only from the line 8-13 due to the action in $t+1$. Then the last action at $t+2$ that changes the substation 3 further disperses electricity from the substation 4. In the end, a more balanced distribution is achieved.

## A.5 ACTIVATION OF THE AGENT

As we mentioned in subsection 3.1, the agent acts only in hazardous situations, i.e. there is a line of which usage rate (ratio between current flow and thermal limit) is larger than the threshold hyperparameter $\delta_h$. Note that the usage rate larger than 1.0 implies that a line is overflowed.

Table 5 shows how the final performance changes according to $\delta_h$ in the test scenarios. If $\delta_h$ is too high, e.g. $\delta_h = 1.1$, the agent may not be able to recover from the hazardous situation, and show relatively worse performance. On the other hand, the agent with $\delta_h = 0.8$ faces more diverse situations, requiring far more samples to reach the performance of the other agents with higher $\delta_h$.

