# OpenReview forum: "Winning the L2RPN Challenge: Power Grid Management via Semi-Markov Afterstate Actor-Critic"
_ICLR.cc/2021/Conference — ICLR 2021 Spotlight_

### Official Review · AnonReviewer3 · 2020-10-28
**Great work**

**Rating:** 9
**Confidence:** 4

**Review:**

Pros:

- The paper presents an RL approach to train a model which may control a power grid. In that effort another actor-critic method coined as SMAAC is suggested which may become useful in other applications as well.
- The authors used the concept of afterstates to reduce the huge state-space offered by L2RPN problem. They used Graph Neural networks along with the transformer based attention mechanism to achieve state of the art performance.
- Formulating the hierarchical  decision model is very beneficial as it allows the exploration without violating the grid boundaries.
- It is shown using the case study that the concept of afterstate has played crucial rule in the success of the model proposed. The same technique does not show good results with SMAAC\AS, which is  SMAAC without afterstate.

Cons:
- In section 3.1 it is said "Thus, for line switch actions, we simply follow the rule always reconnecting the power lines whenever they get disconnected due to the overflow" but this can also cause severe damage to the appliance. There is not enough detail provided on this, but I believe that re-connection must be done AFTER the grid state is improved so the appliances are not damaged.
- According to Kundur 1994, a system goes through "Alert state" and "Emergency state" from normal state to get into "In extremis state".  The desired behavior would be to do restorative action in the "Alert state" to avoid any damage, if not possible then in "Emergency state". But authors have chosen to act only in hazardous situations (section 3.1). A little more discussion would be beneficial in this regard. Apparently, it does not seem difficult to act sooner, as one only has to reduce the width of allowed boundaries, but does this have any effect on exploration? that is not clearly discussed in the article.
- There is a small spelling mistake in section 5.1 "largest" is written as "largeest"

---

> ### Author Response · Authors · 2020-11-17
> **Response to Review #3**
>
> Thank you very much for your constructive feedback.
>
> **[Always reconnection rule]** Considering the real-world where overflow can cause severe damage to the appliance, we also agree that the power line must be reconnected only when there is no overflow. However, the severe damage was not implemented  in the official L2RPN WCCI 2020 simulator. We believe that this will be naturally addressed by reflecting it as a penalty in the simulator.
>
> **["Alert state" and "Emergency state" & Changing threshold of hazard]** We were not familiar with the Kundur 1994 reference when we were working on the problem, and we just followed the prior practice used by the previous competition winners in defining the hazardous state. Thus in our case, the hazardous state would be more like the union of alert states and emergency states.
> We are currently conducting an additional set of experiments that analyzes the effect of threshold conditions for the hazardous state, and we will update the paper with these results as soon as they are ready. We expect that the updated paper will be uploaded within the next 4~5 days.

---

> > ### Author Response · Authors · 2020-11-23
> > **Response to Review #3**
> >
> > **[Changing threshold of hazard]**
> > We added the experiment of the different threshold conditions in Appendix A.5.

---

### Official Review · AnonReviewer2 · 2020-10-28
**Interesting applied work; please mention sustainability in the paper**

**Rating:** 7
**Confidence:** 4

**Review:**

This paper proposes an effective method for managing power grid topology to increase efficiency. They use Transformer attention over a Graph Neural Network as the basic architecture, then propose a hierarchical technique in which the upper level learns to output goal network topologies, which are then implemented by a lower-level policy or a rule-based algorithm. An ablation study reveals that one of the most important components of the algorithm is using an "afterstate" representation, which learns a value function for the state after the agent changes the topology, but before the network is affected by random external factors, including supply and demand.

Strengths:
- The paper tackles a difficult, high-dimensional, real-world problem.
- The architecture chosen is sophisticated and appropriate to the problem at hand.
- The approach won the L2RPN WCCI 2020 competition, showing its effectiveness above other approaches in terms of both significant performance gains, and a significant reduction in computational complexity.
- The experiments benchmark against a thorough set of baselines that use the same architecture, showing additional benefits for the proposed afterstate/topological approach.
- The idea of the afterstate representation is interesting, and the authors provide an ablation study demonstrating its importance.
- The discussion in Section 5.2 about why DDQN and SAC perform worse than a no-op policy was interesting.

Weaknesses:
- I was disappointed that this paper does not mention that an important benefit of increasing the efficiency of power grids through automation is to reduce energy consumption, and thereby reduce reliance on carbon intensive sources of energy, improve sustainability, and help fight climate change. These goals are mentioned in the description of the L2RPN WCCI challenge: https://competitions.codalab.org/competitions/24902, and are further discussed in https://arxiv.org/pdf/1906.05433.pdf. Instead of mentioning sustainability, the intro opens with "The power grid [...] has become an essential component of modern society". Because this paper describes the winning algorithm for the competition, it has the potential to be impactful. The authors should mention this potential benefit of their work, as it could help motivate other AI researchers to work on this important problem. If the authors can include a reference to the potential for increasing sustainability through better grid management, I would increase my score.
- The clarity of the paper can be improved. For example, the intro talks about the difference between a state-action pair and an afterstate, but it is not apparent by that point in the paper what the difference is. Referencing topology vs. random external factors would be helpful here. Similarly, the terminology in Section 3.2 could be modified to improve clarity, perhaps by putting more emphasis on the fact that the paper essentially transforms the action space and learns a value function over topologies, rather than low-level actions.
- More justification for the need for the reparameterization trick could be provided.

---

> ### Author Response · Authors · 2020-11-17
> **Response to Review #2**
>
> Thank you for your constructive feedback.
>
> **[Potential benefit of increasing the efficiency of power grids]** Thank you for your helpful suggestion. We reflected your point in the first paragraph of the introduction. However, please understand that the L2RPN WCCI testbed only contains a small portion of sustainable power sources. Thus, we didn’t put too much emphasis on sustainability.
>
> **[Clarity in introduction and section 3.2]** We add additional explanation for clarification in the introduction and section 3.2 as suggested.
>
> **[Justifying reparameterization trick]** A gradient estimator through reparameterization trick is known to be lower variance than the likelihood ratio gradient estimator, resulting in stable learning that leads to higher final performance [1]. We added the justification in Section 3.2.
>
> [1] Tuomas Haarnoja, Aurick Zhou, Pieter Abbeel, and Sergey Levine.   Soft actor-critic:  Off-policy maximum entropy  deep  reinforcement  learning  with  a  stochastic  actor.    In  Jennifer  Dy  and Andreas Krause (eds.), Proceedings of the 35th International Conference on Machine Learning, volume 80 of Proceedings of Machine Learning Research, pp. 1861–1870, Stockholmsmässan, Stockholm Sweden, 10–15 Jul 2018a. PMLR

---

> > ### Comment · AnonReviewer2 · 2020-11-25
> > **Thanks for updating.**
> >
> > Thank you for updating the paper, I have read the updates and they seem appropriate.

---

### Official Review · AnonReviewer1 · 2020-11-03
**A good application of RL showing value of a novel representation**

**Rating:** 7
**Confidence:** 3

**Review:**

This paper reports an application of RL to a power station management problem. Notably this solution won a recent competition organized around this theme. More interestingly, it involves what seems to be an elegant representation that combines 2 ideas (afterstates and smdps) in a compelling way that I expect to have impact for other hierarchical RL problems.

Pros
---
-> Introduces a representation for a power grid management problem that I found elegant and appropriate for the problem. It has 2 parts: Use afterstates as many actions on the power network lead to the same afterstate by graph isomorphism. The second is the use of a semi-MDP. This combination seems to me to potentially be greater than the sum of its parts as the authors only partially discuss: The combinatorial explosion of the action space that sMDP's introduce are controlled by the afterstates.

-> It won the L2RPN challenge, comparisons against standard algorithms like DDQN shows large gain.
-> Exposition is mostly pretty clear.  I would have emphasized and developed more on the point I made in Pro#1 more.

Cons
---
-> Curiously performance of other competitors were not presented.

-> It is not entirely clear to me that the representational innovation (combining sMDPs+ above is in fact original. The authors seem somewhat timid in claiming it. Clearly delineating what the closest related work on this approach is would have helped. Perhaps authors can clarify in rebuttal.

Section-wise comments
---
2.1: Given that this is an external competition, there is a presumption that this is a realistic/important problem setting. A full discussion of the connection to "real" power grid management is out of scope, but maybe a few cites on this topic would be helpful.

   Also reading this paper, one wonders whether it could simply be generalized to a problem in dynamic graph routing i.e. nothing specific in the model about *power*

3.1:
   A little weirdness here where you say that the agent is designed to act in only hazardous situations which means its goal is roughly to keep the load ratio below 1. But the definition of reward would seem to encourage driving efficiency as high as possible, not just preventing hazard.


sec 4:
   as mentioned above, it would be useful (especially for ICLR audience) tot broaden related work beyond power management. RL on graph problems, other ideas to combine afterstates with heirarchical ?

Table 2: Advisable to mention the metric used in the caption of the table, to make skimming the paper easy.

5.2: The comparison to regular sMDP is very important, and drives home the value of afterstates as being crucial to making the heirarchical representation work.

5.3: RAND seems like a bad name, since the order is randomized a priori than kept fixed, maybe call it FIXED?
  Any analysis of why OPTI could not beat DESC?

---

> ### Author Response · Authors · 2020-11-17
> **Response to Review #1**
>
> Thank you for your constructive feedback.
>
> **[Other competitors]** Currently, only the implementation of the 3rd placed participant in the L2RPN WCCI competition is publicly available. We are currently evaluating their method to plot the learning curve (Fig 3) and expecting to finish all the computation within the next 4~5 days. We will upload revised paper including this result as soon as it is ready.
>
> **[Combining sMDPs+afterstate]** We have now added some relevant prior work on SMDPs and afterstates in the related work section.Yet, to the best of our knowledge, the combination of SMDP and afterstate is a novel contribution.
>
> **[Connection to real power grid]** We added a citation about the details of the Grid2Op simulator and its potential connection to real power grid management. (the first paragraph in secion 2.1)
>
> **[Comparison to Dynamic graph routing]** As you pointed out, our method could be potentially applied to general dynamic graph routing problems such as vehicle routing problem or packet routing problem. This remains an important direction for future work.
>
> **[Weirdness in Reward]** Although the agent is only activated in hazardous situations, the reward function drives the agent to seek a goal topology that is safe as well as efficient. It may be possible to further improve efficiency by acting in non-hazardous situations as well, but it was mostly impossible to optimize such an agent in a large scale grid. In fact, the last winner of the competition also used a similar approach, activating the agent only in hazardous situations.
>
> **[Related work beyond power management]** We added a number of other related studies beyond the topic of power management to the related works section.
>
> **[Metric in the caption]** Thanks for your advice. We added the description of the metric used in the caption.
>
> **[Rand name & Analysis of DESC]** As suggested, we modified the name of the corresponding low-level rule design from RAND to FIXED.
> As shown in Figure 4, the performances of all low-level policies are on par except for FIXED (RAND), and statistically indistinguishable. We suspect that it is because SMAAC is resilient to the choice of suboptimal low-level rules: the high-level policy is trained while the low-level policy is fixed to be CAPA, DESC, or FIXED. Regarding the performance of OPTI, we can only suspect that any reasonable low-level policy is good enough for optimizing the high-level policy. We added it in section 5.3.

---

> > ### Author Response · Authors · 2020-11-23
> > **Response to Review #1**
> >
> > **[Other competitors]**
> > We added the algorithm used by the 3rd placed participant in the experiment section.

---

### Official Review · AnonReviewer5 · 2020-11-05
**A good paper with a very narrow scope**

**Rating:** 7
**Confidence:** 2

**Review:**

Summary of the paper:
The paper introduced a new architecture and a new hierarchical reinforcement learning approach for the power grid control task (in form of the Power Network Challenge L2RPN). The paper starts with outlining the challenges of the problem, statespace size, action-set size and exploration issues. It then introduces the "afterstate" abstraction, which the authors argue is useful for the task because it should be more compact than the "true" MDP state representation. They then define how they model the resulting (semi) MDP, and how they adapt a Soft Actor Critic approach to their proposed hierarchical solution. In particular, their solution introduces a hierarchy in the policy space, where the higher level action is a representation of the desired power grid configuration, and the lower actions are the actual sequence of actions to achieve the desired outcome. The authors also argue that such a hierarchy helps with exploration, because exploration can mostly happen on the higher level only, reducing the explorative complexity. The paper then continues by introducing the function approximator used - a graph neural network with a transformer as the GNN block. The paper finishes by showing that the proposed approach outperforms several baselines by a big margin.

Commentary on the goal of the paper:

The paper is highly application focused on very domain specific. However, given the overall relevance of the application - more efficient and sustainable power grids, I think that this limit in scope is perfectly justifiable.

Strengths:
- The paper clearly lays out the different challenges of the domain
- the paper offers a well-justified, principled solution to each of the different challenges
- the division of the action space in goal state and realization is highly appealing and could find application in other RL domains
- the results are very strong
- the authors show that their approach outperforms several others in an open benchmark

Weaknesses:
- the paper is a little difficult to follow if the reader is not intimately familiar with the domain; several key concepts of the domain are only explained late in the paper, but referenced quite early (in parcicular, it would be helpful to have a more thorough explanation of "topology control" earlier in the paper)
- the paper has some minor editorial issues (typos; grammar; some citations are missing, in particular for soft actor Critic)
- the paper convincingly argues that the problem should be modeled as an RL problem; since there are no other existing RL solutions to this domain, they create RL baselines that have not been explored in the literature. I find this misleading, as it suggests a "strawman" baseline. Given that the authors claim that their approach outperforms a supervised baseline in the competition, it would have been nice to have an SL baseline that is used to illustrate the differences between the two approaches.

---

> ### Author Response · Authors · 2020-11-17
> **Response to Review #5**
>
> Thank you for your constructive feedback.
>
> **[More Explanation on Topology Control]** As the reviewer suggested, we added a bit more detail on the topology control in power networks into the third paragraph of the introduction section, motivating how the topology control can optimize the efficiency of the power delivery.
>
> **[Minor editorial issues]** Thanks for pointing out the typos, grammar, and missing citations. We have updated the manuscript reflecting the corrections.
>
> **[SL baseline]** There seems to be a misunderstanding due to our bad word choice, and we clarified it in the main text. The winner of the last L2RPN 2019 competition pre-trained the RL agent before interacting online with the environment, which is why we used the term “supervised learning” for the pre-training. The detailed process of the dataset collection used for pre-training is not made public, so we cannot reproduce the baseline agent. However, as far as we understand, their pre-training method is not scalable to the grid used in the L2RPN WCCI competition with # of actions ~= 65000 since their method requires simulating all actions in a large number of states. For your reference, L2RPN 2019 competition involved only about 200 actions.

---

### Public Comment · ~Jianhong_Wang1 · 2020-11-16
**Good Paper but may lack some references in the field of Power Systems**

Dear Authors and Reviewers,


I'm a half-expert who lies between RL, MARL and power flow control. I read this paper, and the general idea is cool, however, it is unfair to miss lots of previous attempts in the field of power systems by applying RL on grid control. This may mislead the guys in the field of machine learning, since applying RL and MARL to power system control is not a new story.

some example references:

[1] Diao, R., Wang, Z., Shi, D., Chang, Q., Duan, J., & Zhang, X. (2019, August). Autonomous voltage control for grid operation using deep reinforcement learning. In 2019 IEEE Power & Energy Society General Meeting (PESGM) (pp. 1-5). IEEE.

[2] Ernst, D., Glavic, M., & Wehenkel, L. (2004). Power systems stability control: reinforcement learning framework. IEEE transactions on power systems, 19(1), 427-435.

[3] Vlachogiannis, J. G., & Hatziargyriou, N. D. (2004). Reinforcement learning for reactive power control. IEEE transactions on power systems, 19(3), 1317-1325.

[4] Yang, Q., Wang, G., Sadeghi, A., Giannakis, G. B., & Sun, J. (2019). Two-timescale voltage control in distribution grids using deep reinforcement learning. IEEE Transactions on Smart Grid, 11(3), 2313-2323.

[5] Hua, H., Qin, Y., Hao, C., & Cao, J. (2019). Optimal energy management strategies for energy Internet via deep reinforcement learning approach. Applied Energy, 239, 598-609.

[6] Huang, Q., Huang, R., Hao, W., Tan, J., Fan, R., & Huang, Z. (2019). Adaptive power system emergency control using deep reinforcement learning. IEEE Transactions on Smart Grid, 11(2), 1171-1182.

[7] Li, F. D., Wu, M., He, Y., & Chen, X. (2012). Optimal control in microgrid using multi-agent reinforcement learning. ISA transactions, 51(6), 743-751.

[8] Dimeas, A. L., & Hatziargyriou, N. D. (2010, July). Multi-agent reinforcement learning for microgrids. In IEEE PES General Meeting (pp. 1-8). IEEE.

[9] Wu, J., Wei, Z., Liu, K., Quan, Z., & Li, Y. (2020). Battery-involved Energy Management for Hybrid Electric Bus Based on Expert-assistance Deep Deterministic Policy Gradient Algorithm. IEEE Transactions on Vehicular Technology.

[10] Wang, W., Yu, N., Gao, Y., & Shi, J. (2019). Safe off-policy deep reinforcement learning algorithm for volt-var control in power distribution systems. IEEE Transactions on Smart Grid.

These are just the example references, and there are so many that need to be explored.


Best Regards,

Jianhong

---

> ### Author Response · Authors · 2020-11-17
> **Response to Jianhong Wang**
>
> Thank you for the references on prior work that we potentially missed as related work. We have included some of the suggested references in the revised paper. However, please note that although the references are about applying RL to power grid management in general, none of them is closely related to our problem, i.e. topology control. To the best of our knowledge, there is no prior literature on applying RL to large-scale power network topology control except the following additional paper we recently found:
>
> Han, Control Method of Buses and Lines Using Reinforcement Learning for Short Circuit Current Reduction, Sustainability, 2020, 12(22)

---

### Author Response · Authors · 2020-11-24
**Summary of the updates in the revision**

We appreciate the reviewers' detailed and careful feedbacks. Based on the feedbacks, we have revised our draft by incorporating the following changes:

* We have added additional explanation about the potential benefit for power grid management through automation, the topology control and the afterstate in the introduction, as suggested by Reviewer 2 and Reviewer 5.
* We have added a citation about the Grid2Op simulator and its potential connection to real power grid management in section 2.1, as suggested by Reviewer 1.
* We have added a bit more explanation of the afterstate value function and justification of using the reparameterization trick in section 3.2, as suggested by Reviewer 2.
* We have included related works about afterstates, hierarchical RL, and RL on graph in section 4, as suggested by Reviewer 1.
* We have included the comparison between the 3rd placed participant of the competition in section 5, as suggested by Reviewer 1.
* We added more explanation about section 5.3, as suggested by Reviewer 1.
* We have run the experiments for the different threshold of hazardous situation in Appendix A.5, as suggested by Reviewer 3.

---

### Decision · Program_Chairs · 2021-01-07
**Final Decision**

**Decision:**

Accept (Spotlight)

**Comment:**

With reviewer scores of (7, 7, 9, 7), and with only one low-confidence score (R5's score of 7 with confidence of 2) it is obvious that the paper should be accepted.